Na+/Ca2+ selectivity in the bacterial voltage-gated sodium channel NavAb

Corry Ben ben.corry@anu.edu.au
Research School of Biology , The Australian National University , Acton , Australia
Separovic Frances
Electronic publication date: 2013 Feb 12
Publication date: 2013
Volume: 1
Electronic Location ID: e16
Received 2012 Nov 30; Accepted 2013 Jan 4
Copyright: © 2013 Corry
Copyright year: 2013
Copyright holder: Corry
License: This is an open access article distributed under the terms of the Creative Commons Attribution License, which permits unrestricted use, distribution, and reproduction in any medium, provided the original author and source are credited.
License URL: https://creativecommons.org/licenses/by/3.0/

Keywords: Ion channel, Ion selectivity, Molecular dynamics, Sodium channel, Action potential, Bacterial channel, Calcium channel, Simulation

Funding: Pawsey Centre Project Merit Allocation Scheme of the NCI This work was supported by computer time from the Pawsey Centre Project in Western Australia and through an award under the Merit Allocation Scheme of the NCI facility at the ANU. The funders had no role in study design, data collection and analysis, decision to publish, or preparation of the manuscript.

==============================
The recent publication of a number of high resolution bacterial voltage-gated sodium channel structures has opened the door for the mechanisms employed by these channels to distinguish between ions to be elucidated. The way these channels select between Na+ and K+ has been investigated in computational studies, but the selectivity for Na+ over Ca2+ has not yet been studied in this way. Here we use molecular dynamics simulations to calculate the energetics of Na+ and Ca2+ transport through the channel. Single ion profiles show that Ca2+ experiences a large barrier midway through the selectivity filter that is not seen by Na+. This barrier is caused by the need for Ca2+ to partly dehydrate to pass through this region and the lack of compensating interactions with the protein. Multi-ion profiles show that ions can pass each other in the channel, which is why the presence of Ca2+ does not block Na+ conduction despite binding more strongly in the pore.

Introduction

Voltage-gated sodium channels create the upstroke of action potentials in excitable cells by opening a Na+ selective transmembrane pore in response to small depolarising signals. (Catterall, 2000; Goldin, 2001; Hille, 2001) These channels are able to rapidly move Na+ ions down their electrochemical gradient, while limiting the passage of other ions such as K+, Ca2+ and Cl−. Until recently, gaining a detailed understanding of the mechanisms leading to ion selectivity in these channels has been difficult due to a lack of structural information. The publication of atomic resolution structures of a number of voltage-gated sodium channels from bacteria in the last 18 months (McCusker et al., 2012; Payandeh et al., 2012; Payandeh et al., 2011; Zhang et al., 2012) has, however, now made it possible to elucidate the physical mechanisms employed to generate sodium selectivity.

Unlike eukaryotic sodium channels that are formed from a single protein chain, the bacterial sodium channels are made up of four identical subunits that surround a central pore. The pore shape is similar to the familiar structure of potassium channels. One set of transmembrane helices form an inverted tepee that lines the inner half of the pore and contains the activation gate. A re-entrant loop between two short helices forms a narrow ‘selectivity filter’ that is wider and shorter than that of potassium channels, as well as being partly lined by amino acid side chains. In these bacterial sodium channels, four identical subunits line the pore and the external end of the filter is surrounded by side chains of four glutamate residues (Glu177), in the same sequence position as the characteristic DEKA ring found in eukaryotic sodium channels (Payandeh et al., 2011) which is known from mutation studies to be essential in creating Na+ selectivity (Favre, Moczydlowski & Schild, 1996; Heinemann et al., 1992; Sun et al., 1997).

The question of how the bacterial sodium channels can discriminate between Na+ and K+ has been addressed in studies based upon the crystal structure of the channel from Arcobacter butzleri (NavAb) (Payandeh et al., 2011). It was initially suggested that the high field strength of the glutamate residues would create an ion binding site more favourable for Na+ than K+ (Payandeh et al., 2011). Two computational studies, however, highlighted the importance of a narrow region just beyond this binding site for creating selectivity (Corry & Thomas, 2012; Furini & Domene, 2012). We have suggested that the discrimination at this point arises from the inability of K+ to fit through the narrow region of the pore along with surrounding water molecules in the optimal geometry (Corry & Thomas, 2012).

How ion channels discriminate between Na+ and Ca2+ is also not straight forward to explain. The family of voltage-gated sodium channels is closely related to the voltage-gated calcium channels. They share a similar topology comprising four homologous domains each containing 6 transmembrane sections. Furthermore, both classes of channel are believed to share some structural features in the selectivity filter (Hille, 2001). Mutation of a few key residues in both eukaryotic and bacterial sodium channels can decrease or reverse Na+ selectivity to create a pore with characteristics similar to the voltage-gated calcium channels (Heinemann et al., 1992; Shaya et al., 2011; Sun et al., 1997; Yue et al., 2002). Similarly, substitutions in the conserved ‘EEEE locus’ of calcium selective channels can allow Na+ permeation or remove Ca2+ currents (Cibulsky & Sather, 2000; Ellinor et al., 1995; Tang et al., 1993). Thus the difference in selectivity seen in sodium and calcium channels appears to be due to only minor differences in the protein.

A number of computational studies made in the absence of atomic resolution information have examined the nature of selection between Na+ and Ca2+ (Boda et al., 2009; Corry et al., 2000; Corry et al., 2001; Corry & Chung, 2006; Corry, Vora & Chung, 2005; Gillespie, 2008; Nonner, Catacuzzeno & Eisenberg, 2000; Vora, Corry & Chung, 2005; Yang, Henderson & Busath, 2004). It has been suggested, for example, that the electrostatic interactions that bind Ca2+ and Na+ differently (Corry et al., 2001; Corry & Chung, 2006) or the simple competition between differently charged ions for a limited space (Nonner, Catacuzzeno & Eisenberg, 2000) can be enough to explain the different behaviours of sodium and calcium channels, but these ideas are yet to be tested in models containing atomic detail. Experimental studies have shown that eukaryotic calcium channels are continually occupied by at least one Ca2+ that guards against permeation of other ions (Hess & Tsien, 1984).

How NavAb distinguishes between Na+ and Ca2+ has not yet been explained, and this manuscript aims to address this issue. Na+currents are more than 20 times greater than Ca2+ currents in NavAb when measured without the other ion present (Payandeh et al., 2011). Reversal potentials of the closely related NachBac and NavSp1 channels measured in asymmetric solutions show a preference for Na+ over Ca2+ of 7–15 (Shaya et al., 2011; Yue et al., 2002). These two ions are very similar in size, yet differ in charge, and so the mechanism of selectivity must rely of the consequences of this charge difference. Further confusing this issue is that a ring of four glutamate residues has been found to be essential to create Ca2+ over Na+ selectivity in calcium channels, so one can ask why the presence of the ring of glutamates present in the bacterial sodium channels does not yield Ca2+ selectivity. Here we employ the recent structural information in molecular dynamics simulations to try to understand why the passage of Na+ is preferred over Ca2+ in the bacterial voltage-gated sodium channel NavAB.

Materials and Methods

Protein coordinates were obtained from the protein database, PDB accession code 3RVY (Payandeh et al., 2011). Only the pore forming region from residue 115 to 221 was included in the simulations. In order to conduct the molecular dynamics simulations, the protein was placed in a pre-equilibrated POPC lipid bilayer, solvated in a 72 × 72 × 82 Å box of TIP3P water with 250 mM NaCl, yielding a system with approximately 41,000 atoms as depicted in Fig. 1. The protein was initially held fixed while the lipid and water were allowed to equilibrate for 2 ns, before the protein alpha carbons were restrained with a harmonic potential of force constant decreasing from 10 to 0.01 kcal/mol in 8 steps each lasting 1 ns. To account for the lipid molecules protruding into the centre of the pore in the crystal structure, the tails of four lipid molecules were restrained to the crystallographic positions with a force constant slowly increasing during the period in which the protein was restrained. During data collection they were restrained with a force constant of 0.1 kcal/mol/Å2. A final 15 ns of equilibration were conducted at the final restraint values. In longer unrestrained simulations not described here, lipid remains constantly present in the lateral phenestrations. For the simulations involving Ca2+, one Na+ was replaced with Ca2+ and an additional Cl− added to neutralise the system.

Figure 1 Simulation system.

A cross section of the simulation system is shown, with two of the four protein chains shown in purple, lipid in brown, Na+ in yellow, Cl− in green and the volume sampled by water molecules indicated by the transparent surface.

Potentials of mean force (PMF) were calculated using umbrella sampling (Torrie & Valleau, 1974). To ensure other ions did not enter the selectivity filter during these simulations, all other ions were kept out of a sphere of radius 13 Å centred on the middle of the selectivity filter using harmonic repulsive potential with force constant 40 kcal/mol/Å2. When making 1 ion PMFs, the axial separation (z) of the ion of interest and the centre of mass of the alpha carbons of the selectivity filter (residues 175–178) was restrained in 0.25 Å steps in the z-direction with a force constant of 12 kcal/mol/Å2. For the two ion PMFs a force constant of 7 kcal/mol/Å2 and 0.5 Å steps were employed. Simulations lasting 2 ns were conducted at each window, with the first ns regarded as equilibration time and analysis conducted on only the second ns. Thus each single ion PMF involves 101 simulations for a total of 202 ns. The 2 ion Ca2+/Na+ PMF involves 695 simulations for a total of 1390 ns. To reduce the amount of equilibration time required in each umbrella window, a number of starting coordinates were generated in which one or two ions were swapped with equilibrated water molecules throughout the length of the pore. In the two ion case, this meant starting coordinates were generated with the ions separated by integer numbers of water molecules. Collective analysis was made with the weighted histogram analysis method. Kumar et al. (1992) using the implementation of Grossfield (Grossfield). The axial coordinate used in all the graphs is zeroed at the position of the centre of mass of the alpha carbons of the selectivity filter. The single Ca2+ PMF was obtained three times from three independent sets of simulations to gain an idea of the reproducibility of the results.

Unless otherwise stated, all simulations employed the CHARMM27 force field with CMAP correction for the protein and Ca2+, (MacKerell et al., 1998) CHARMM36 for lipids, (Klauda et al., 2010) and Na+ and Cl− parameters from Joung (Joung & Cheatham, 2008). Because of the uncertainties involved in using a non-polarisable force field to describe interactions with Ca2+, care should be taken not to overemphasise quantitative results involving this ion. Simulations were conducted with NAMD (Phillips et al., 2005) with periodic boundary conditions using the particle mesh Ewald scheme for calculating electrostatic interactions, a 1 fs time-step at constant pressure (1 atm) and temperature (298 K) (NPT ensemble). A harmonic potential of 0.01 kcal/mol/Å2 was held on the alpha-carbon atoms, except for those forming the selectivity filter (residues 174–183) during data production. Coordination numbers were defined as the number of non-hydrogen atoms within 3.0 Å of the ion, the approximate position of the first minimum in the bulk ion–water oxygen radial distribution function.

To calculate the dehydration energies of the ions, we used the method of alchemical free energy perturbation (FEP) following the approach previously taken to examine partial dehydration of monovalent ions (Song & Corry, 2009). To create a dehydration scenario similar to that for ions entering the channel, we first generated a water sphere of radius 15 Å with the test ion fixed at the center to represent the bulk solution. Beside this (20 Å from the center of the large sphere) we placed a smaller water sphere of radius 3.2 Å that contained the desired number of water molecules. Large constraints (100 kcal/mol Å2) were applied to keep all of the atoms in the constant-volume spheres, as was done previously (Thomas, Jayatilaka & Corry, 2007). Then, alchemical FEP calculations were performed for these systems in which the test ion was moved from the bulk–water sphere to the small water cluster, replacing a water molecule that was moved in the opposite direction. All simulation parameters except for the force field parameters for Na+ and Ca2+ are the same as our previous study (Song & Corry, 2009). To calculate the cost of removing water beyond the second solvation shell a similar procedure was conducted except, the small sphere had a radius of 6 Å and contained 32 water molecules, the average number found within the first and second solvation shells for both ions in bulk simulations. Each FEP calculation was repeated 9 times and the results show the average free energy change and the standard error in the mean.

Results

The single ion PMFs for Na+ and Ca2+ depicted in Fig. 2 show that both ions are attracted from bulk (at the right side of the figure) into the selectivity filter of the channel and have an energy minimum adjacent to the ring of Glu177 residues at around z = 5 Å. In this position each ion is directly coordinated by the side chains of one glutamate residue as well as that of Ser178 as depicted in Figs. 3A and 3C and evident in a plot of ion coordination numbers (Fig. 4). Ca2+ binds more strongly to the protein than Na+ as evidenced by the greater difference in the free energy of the binding site relative to bulk. Unlike in previous MD studies of a model pore containing four glutamate residues in which the Ca2+ binds centrally coordinated by multiple carboxylate groups (Yang, Henderson & Busath, 2004), here the ion directly coordinates to only one glu side chain. Part of the reason for this is likely to be that each of the glu side chains is held in position by hydrogen bonds to Ser 1180 and Met 1181 on the helix further from the pore.

Figure 2 Single ion potential of mean force.

Single ion potential of mean force (PMF) for Na+(blue), Ca2+(red) and K+(black dashed line) is shown as a function of the axial position of the ion, zeroed at the centre of the selectivity filter. The ion is in bulk water at the right side of the graph and in the central cavity of the channel on the left hand side. For Ca2+ the PMF shown is the average of 3 independent sets of simulations, with the standard deviation in the three values shown shaded in grey.

To pass further into the pore, an ion must leave the embrace of the glutamate side chains. We have previously suggested that it is the unfavourable location where the ion passes through the plane of these side chains at which selectivity for Na+ over K+ arises (Corry & Thomas, 2012). As for Na+ and K+, Ca2+ experiences an energy barrier at this position.

In contrast to both Na+ and K+, Ca2+ experiences a large energy barrier further into the pore, at around z = 0.5 Å in Fig. 2, which it must overcome to enter the central cavity of the pore. The size of the maximum energy barrier seen by each ion type (Na+ < K+ < Ca2+) is in accord with the magnitude of current measured for each ion (Na+ > K+ > Ca2+) (Payandeh et al., 2011). The energy change for an ion to move from the energy minimum in the filter to the start of the central cavity is about 1 kcal/mol for Na+ and 3 kcal/mol for Ca2+, reinforcing that Ca2+ binds more strongly in the filter than Na+. The single Ca2+ PMF was obtained 3 times from 3 independent sets of simulations to gain an appreciation of the reproducibility of the results. In all cases the maximum barrier for Ca2+ permeation occurs at the same location near z = 0.5 Å.

To conduct our simulations we use the ‘pre-open’ structure of the channel, in which the selectivity filter is believed to be in an open state, but the activation gate is closed at the intracellular end of the pore (Payandeh et al., 2011). As a consequence, a large increase in free energy is seen at the left hand end of the PMFs shown in Fig. 2 where the ion approaches the closed gate, and energy values in this region cannot be expected to represent those in a fully open channel. While it is possible that the closed gate could alter the energy profiles in the filter, we expect that the water filled cavity to largely screen the effect of the gate on the energy values in the filter.

To understand the reason for the large energetic barrier seen by Ca2+, we show a snapshot of either Na+ or Ca2+ in this position in Figs. 3B and 3D. Here, the ion has moved beyond the Glu177 side chains, but not so far into the pore as to be surrounded by the backbone carbonyls of Thr175 and Leu176 which have been suggested to form an ideal environment for a solvated ion (Payandeh et al., 2011). Rather, the ion is surrounded by 5 or 6 water molecules which can only form limited interactions with the protein. Inspection of the coordination numbers of the ions in Fig. 4 shows that Na+ maintains a coordination number close to 6 as it passes through this part of the pore, and begins to form occasional contacts with the backbone carbonyl groups. The coordination number of Ca2+, however, shows 2 regions at which it drops below the average value of 7. When adjacent to the Glu177 residues, the coordination number begins to drop (labelled ① in the figure), but the strong interaction with the charged residues means this remains a favourable location for the ion. There is another significant drop at the location of the energy barrier (labelled ②) to a value less than 6.5.

Figure 3 Snapshots of ions in the channel.

Simulation snapshots showing the ion and coordinating water when either Na+ (A) or Ca2+ (C) is at the external binding site, or when the Na+ (B) and Ca2+ (D) is at the location of the largest barrier experienced by Ca2+.

Figure 4 Coordination numbers of ions in the channel.

Coordination numbers for (A) Na+ and (B) Ca2+ as a function of the axial position of the ion in the pore. The total coordination number is shown in black, while the contribution from water (blue), glutamate side chains (red) and other protein residues (green) are indicated. Standard errors in the mean are smaller than the data points and are not shown.

In Table 1, we show the energetic cost of removing water from the hydration shell of Na+ and Ca2+ to see if this can account for the energy barriers seen in our PMFs. For Na+, the penalty for removing a single water from the inner hydration shell costs ∼3  kcal/mol. In contrast, Ca2+ holds on to both its inner and outer solvation shells much more strongly due to the increased electrostatic interactions of this ion with water molecules. From Table 1 it can be seen that removing one water molecule from the inner shell of Ca2+ (going from a coordination number of 7 to 6) has an energy cost of ∼8  kcal/mol. This suggests that removal of (on average) half a water molecule from the solvation shell could account for half the energy barrier seen by Ca2+. The results in table also show that Ca2+ is sensitive to the environment outside the immediate coordination shell and so it is plausible that changes in the first solvation shell and limited interactions with more distant water could account for the barrier seen in our PMFs.

Table 1 Free energy (kcal/mol) required to partly dehydrate each type of ion. Each value represents the energy to remove an ion from bulk to a situation in which it is surrounded either by n water molecules or its first and second solvation shell (column labelled 2nd shell) as determined from FEP simulations. The bracketed values show the standard error in the mean for the last digit calculated from 9 independent simulations.

	n		
	0	1	2	3	4	5	6	7	2nd Shell	
Na+	71.2 (1)	47.2 (2)	32.5 (2)	20.9 (3)	11.9 (3)	6.7 (2)	4.0 (3)	N/A	−0.1 (0.2)	
Ca2+	312.2 (6)	247.8 (9)	200 (1)	161.6 (6)	123.6 (7)	95 (1)	71.4 (6)	63.6 (7)	11.7 (7)	

To further support our claim that the differences in coordination numbers can account for the different free energy barriers seen by the ions in the channel, we plot the average ion–water and ion–protein interaction energy as a function of ion position for Ca2+ in the selectivity filter in Fig. 5A. The ion–protein and ion–water interactions are almost mirror images of one another. It can be seen that a strong attractive interaction between the ion and protein arises when the ion is close to the ring of Glu177 residues (z = 5 Å). At this point, the replacement of water in the coordination shell with protein yields a less negative ion–water interaction. As the ion moves through the centre of the selectivity filter, the ion–protein interaction becomes less negative and the ion–water interaction more negative. In Fig. 5B we plot the sum of the ion–protein and ion–water interactions as either a single Na+ or Ca2+ moves through the pore (vertically shifted so the results for the two ions can be compared). The fact that the two interaction terms are almost mirror images means that the sum of the two produces a relatively flat line. While the sum of the interactions fluctuates as a function of ion position, it is much more stable for Na+ than for Ca2+. At the location of the energy barrier for Ca2+ seen in the PMFs (z = 0.5 Å), both ions see an upward deflection in the curve, indicating that the ion–protein interaction is not able to compensate for the reduced ion–water interaction that is enforced by the limited space in the pore. Reinforcing the conclusion drawn from the graph of coordination numbers, the barrier seen for Ca2+ seems to be due to a less than ideal combination of ion–protein and ion–water interactions. In other words, Ca2+ experiences a degree of dehydration which is not compensated by interactions with the protein. The second and further hydration shells are also held more strongly by Ca2+ than Na+, and so their removal will also disfavour Ca2+ as this does not appear to be compensated by interactions with the protein.

Figure 5 Interaction energies of the ions with water and protein when in the channel.

Ion–water and ion–protein interaction energies. (A) the Ca2+–protein (black) and Ca2+–water (blue) interaction is plotted as a function of the axial position of the ion in the pore. (B) The sum of the ion–water and ion–protein interaction energies is shown for both Na+ (blue) and Ca2+ (red) as a function of the axial position of the ion in the pore. As only changes in the total energy with position are important for the discussion presented, the curves on B have been vertically shifted to zero at the left hand side to allow for the results for Ca2+ and Na+ to be more easily compared.

Having described the differences in the permeation of a single Na+ and a single Ca2+ ion through the channel, we next turn our attention to what would occur if multiple ions are allowed to enter the pore. In Fig. 6 we plot the PMF as a function of the position of two Na+ ions in the pore. This graph is generated from the same data presented in our previous work, (Corry & Thomas, 2012) and shows how the presence of two Na+ ions can slightly reduce the barriers to ion conduction compared to the case when a single ion permeates on its own. The plot shows that one ion is likely to always occupy the binding site by the Glu177 residues and that conduction is most likely to occur when a second ion displaces the first ion from this position and pushes it through the pore in a loosely coupled knock on mechanism (i.e., following the dotted line from state 1 to 2 to 3 as shown in the insets to the figure). The maximum barrier for this process is approximately 2.5 kcal/mol (Corry & Thomas, 2012). Figure 6 also shows that it is possible for a Na+ ion to bypass a resident ion (crossing the diagonal on the plot).The energy barrier for this process is only 2 kcal/mol greater than that seen for knock on conduction. Thus, ions may be able to pass each other in the pore and so conduction may not be a strictly single file process.

Figure 6 Potential of mean force for two sodium ions in the channel.

The PMF is plotted as a function of the positions of two Na+ ions in the pore. Contours are at 1 kcal/mol intervals. Representative snapshots are shown in the insets for three low energy configurations, whose locations are shown on the plot. The approximate lowest energy pathway for ion permeation is shown by the dotted line.

The PMF obtained with two Ca2+ ions in the channel is plotted in Fig. 7. The barrier for Ca2+ conduction is lowered by the presence of the second ion, as can be seen by the reduced barrier to move along the path indicated by the dotted line compared to moving along the top edge of the figure. The maximum barrier for conduction drops from ∼8 kcal/mol in the single ion PMF to ∼4.5 in the two ion case, more consistent with the observation that Ca2+ currents are 20 fold less than Na+ currents. Thus, Ca2+ will also be expected to conduct in a knock on mechanism. The largest barrier to permeation is located at the same position as for the single ion PMF (z = 0.5 Å), indicating that the analysis of the origin of this barrier is relevant in the case of multi-ion conduction. Figure 7 also indicates that Ca2+ ions are unlikely to pass each other in the pore.

Figure 7 Potential of mean force for two calcium ions in the channel.

Two ion potential of mean force for Ca2+. The PMF is plotted as a function of the positions of two Ca2+ ions in the pore. Contours are at 1 kcal/mol intervals. Representative snapshots are shown in the insets for four low energy configurations, whose locations are shown on the plot. The approximate lowest energy pathway for ion permeation is shown by the dotted line. A smaller range of coordinates is shown compared to the other two ion PMFs as the ions are unlikely to pass each other in the pore.

Figure 8 Potential of mean force for one calcium and one sodium ion in the channel.

Mixed ion potential of mean force. The PMF is plotted as a function of the position of one Ca2+ ion (x-axis) and one Na+ ion (y-axis). Contours are at 1 kcal/mol intervals. Representative snapshots are shown in the insets for six low energy configurations whose locations are shown on the plot. The approximate lowest energy pathway for Na+ to permeate through the pore by passing a resident Ca2+ ion is shown by the dotted line.

In Fig. 8 we show the PMF in a mixture of Ca2+ and Na+. If we start with the Ca2+ adjacent to Glu177 (as depicted in inset 3) then we can see than the barrier for Ca2+ to move through the channel is no smaller with Na+ nearby (moving from the state shown in inset 3 to inset 1) than if Ca2+ is moving on its own (state 3 to state 2). In each case the maximum barrier is ∼6.5  kcal/mol, which is similar to that seen in the single ion Ca2+PMF. The figure also shows the profile for Na+ to bypass Ca2+ (i.e., moving along the dotted line from state 3 to 4 to 5 to 6). Here the maximum barrier of ∼5  kcal/mol arises at the point at which the ions move past one another (between states 4 and 5). Thus, it is more likely for Na+ to bypass Ca2+ than for Ca2+ to be pushed through the pore. Finally we can see that if Na+ starts in the pore (state 5) then Ca2+ can easily displace it and push it through the pore (state 6).

Discussion

In the absence of other cations, our simulations imply that the conduction of Na+ is rapid due to the small energy barriers experienced by these moving through the pore in a sequential knock on process. In contrast, we see a much larger barrier for Ca2+ to pass, in accord with its lower current. The barrier for Ca2+ is slightly lower in the presence of Na+ and is further lowered in the presence of a second Ca2+.

The results shown in Figs. 6 and 8, however, also indicate that ions can pass each other inside the pore. When two Na+ ions are in the pore (Fig. 6) the barrier for ions passing each other is only a little larger than seen for knock on conduction, while Na+ can pass K+ with almost no additional barrier (Corry & Thomas, 2012). If Ca2+ resides in the pore, the entry of a Na+ ion does not increase the likelihood of the Ca2+passing through the pore. Na+ can also move around a resident Ca2+ ion and conduct through the pore, however the barrier for this process is larger than when only Na+ is present. As a consequence, we would expect the presence of Ca2+ in the filter to attenuate Na+ currents, but not to completely prevent them.

It has previously been shown that divalent cations interfere with the conduction of Na+ in eukaryotic sodium channels. (French et al., 1994; Mozhayeva, Naumov & Khodorov, 1982; Ravindran, Schild & Moczydlowski, 1991; Tanguy & Yeh, 1988; Taylor, Armstrong & Bezanilla, 1976; Woodhull, 1973; Yamamoto, Yeh & Narahashi, 1984) But, in the bacterial channel NachBac, the presence of 1 mM Ca2+ only slightly reduces Na+ currents, (Ren et al., 2001) This could only arise if either Ca2+ did not enter the pore or if Na+ can pass Ca2+ ions resident in the pore. Our simulations results would suggest a larger attenuation of Na+ currents in the presence of Ca2+ in NavAb than seen experimentally for NachBac. While it is possible that these channels are different in this respect, the sequence similarity of the two proteins makes this seem unlikely. The proximity of the four Glu177 residues makes it possible that one or more of these residues will be protonated as is suspected to be the case in calcium channels (Corry et al., 2001; Root & Mackinnon, 1994). Previously we suggested that the protonation of a single Glu side chain would not influence the selectivity of the channel for Na+ over K+, (Corry & Thomas, 2012) however, this does reduce the strength of binding of the ion to the external site. The position of the saddle point in Fig. 7 shows that the easiest way for Na+ to pass a resident Ca2+ is for the Ca2+ ion to back slightly out of the binding site. Any reduction in affinity of the site would make it easier for a resident Ca2+ ion to do this. This, in turn, may expediate the passage of Na+ past a resident Ca2+, and thus limit the attenuation of Na+ currents seen in the presence of Ca2+.

Another important factor to consider is how accurately the classical simulations used here can be expected to reproduce the free energies of the transport process. Ideally, the ion parameters need to be able to reproduce the interactions of the ions with a range of molecules (including protein functional groups and water) as well as the structure and dynamics of interacting molecules, something which is difficult to achieve with classical non-polarisable models. Dealing with divalent ions is particularly difficult in non-polar force fields. Due their high charge density, divalent ions are more likely to polarise surrounding molecules than is the case for monovalent ions, (Bako, Hutter & Palinkas, 2002; Bucher & Kuyucak, 2008) an effect which is not considered in the present study. The parameters for Na+ and K+ have been shown to accurately reproduce a range of properties including hydration energies, the structures of ion–water clusters and crystals and binding energies (Joung & Cheatham, 2008). The parameters for Ca2+ were also optimised to reproduce hydration free energies and binding energies with water, (Marchand & Roux, 1998) and have been used in a range of studies of biological molecules, (Cates, Teodoro & Phillips, 2002; Sotomayor & Schulten, 2008; Sotomayor et al., 2012) but have not been tested as rigorously on specific interactions with protein functional groups or structural aspects. Despite these caveats, we believe that the general principles that yield strong binding of Ca2+ to the external binding site and the uncompensated energy cost of dehydration that yields the main energy barrier for Ca2+ in the pore, are likely arise even with a polarisable force field. However, caution should be applied to making quantitative predictions. It is certainly possible that the Ca2+ parameters influence the ease by which Na+ can pass resident Ca2+ ions.

It is interesting to ponder how changes to NavAb could lead to Ca2+ selectivity. A small number of mutations have been shown to be able to convert bacterial sodium channels into being Ca2+ selective (Shaya et al., 2011; Yue et al., 2002). These involve introducing additional acidic residues at the mouth of the channel, the presence of which seems to be essential for generating Ca2+ selectivity. The additional negative charge may increase the chance of a second Ca2+ entering the pore to yield knock on Ca2+ conduction. The PMF determined here with two Ca2+ in the pore suggests that the presence of multiple divalent ions can reduce the major barrier to Ca2+ seen in our simulations, but given that this barrier arises far from the location of the mutations further simulations will be required to shed light on this issue. The additional negative charge may also accentuate the difference in the binding affinity of the filter for Ca2+ compared to Na+.

Given that Ca2+ binds more strongly than Na+ in NavAb, and experimental studies of eukaryotic calcium channels show they are permanently occupied by Ca2+, (Hess & Tsien, 1984) it is also possible that Ca2+ selectivity could be obtained by preventing Na+ from being able to pass a resident Ca2+ ion. Figure 8 indicates that such passing is likely to occur in the channel mouth, slightly external to Glu177. So, selectivity for Ca2+ over Na+ may also be achieved by narrowing the diameter of the pore entrance to create genuinely single file conduction. If such a narrowing took place external to Glu177, it need not significantly change the diameter of the narrow point of the selectivity filter and so could keep this consistent with experimental estimates of the minimum pore diameter in eukaryotic calcium channels (>5.5 Å) (Mccleskey & Almers, 1985). Similarly, if the Glu177 side chains were less constrained in position by hydrogen bonds to the rest of the protein, then it is possible that multiple side chains would coordinate a passing Ca2+ and occlude the pore as is studies of a model pore (Yang, Henderson & Busath, 2004). Site directed mutagenesis and electrophysiology has shown that in calcium channels the four glutamate residues are not symmetrically positioned (Ellinor et al., 1995) and it is possible that this would also weaken the binding affinity of Ca2+ to the filter to allow conduction to occur more easily.

Conclusions

In agreement with experimental data, the single and multi-ion free energy profiles calculated here indicate that the Ca2+ conductance of NavAb is likely to be much smaller than for Na+. While the results presented here signify that the channel will prefer the conduction of Na+ in mixed solutions, the difference in energy barriers for the conduction of Na+ and Ca2+ in mixed solution are small, in agreement with permeability ratios in the order of 10:1 (Yue et al., 2002). The reason that Ca2+ has difficulty in passing through the pore appears to be due to the limited space midway through the selectivity filter between the sections lined by the Glu177 side chains and the carbonyl backbones of Leu176. The lack of both space and polar groups here limits ion coordination numbers without offering compensating ion–protein interactions. Ca2+ suffers a more significant drop in coordination numbers at this point and thus a larger dehydration penalty than does Na+. Our results indicate that ions can pass each other in the pore, which is why Ca2+ does not block Na+ currents, however our data does predict a larger attenuation of Na+ currents by Ca2+ than was seen in the related channel NachBac.

Michael Thomas is thanked for helpful discussions.

Additional Information and Declarations

Competing Interests

Author Contributions

Ben Corry is an Academic Editor for PeerJ.

Ben Corry conceived and designed the experiments, performed the experiments, analyzed the data, contributed reagents/materials/analysis tools, wrote the paper.

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
