# Peer review of "Na+/Ca2+ selectivity in the bacterial voltage-gated sodium channel NavAb"

_PeerJ, doi:10.7717/peerj.16_

## Round 0.1 · original submission · Minor Revisions

Please note the comments about the barrier to Ca transport transport made by Reviewers 1 & 3.

·

Basic reporting

Corry reports his evaluation of the potential of mean force for Na+ and Ca++ transport through the bacterial voltage-gated sodium channel, NavAB, using the transmembrane portions of the protein constrained to the crystal structure in PDB 3RVY. In the crystal the tails of four lipid molecules project into the central cavity, which were retained. The current CHARMM force field is used except for specialized parameters for Na+ and Cl-, with standard molecular dynamics techniques for umbrella sampling. One-ion and two-ion PMFs, ion coordinations numbers, and some decomposition of interaction energies are presented using current standards.

Experimental design

The principle question addressed is: Why does the channel have a higher permeability and conductance for Na+ than for Ca++, especially considering that the selectivity filter has four glutamate side chains, more like a Ca++ channel than a Na+ channel? This is a significant question of current interest. The background and current pertinant literature are well described, except for some neglect of early theory on space-charge competition in the charged selectivity filter (Nonner W, Catacuzzeno L, Eisenberg B. 2000. Binding and selectivity in L-type calcium channels: a mean spherical approximation. Biophys J. 79:1976-92) and of some similar simulations with similar results done by Yan Yang under my supervision using a primitive structure for the channel but very similar glutamate side chain positions (Yang, Y., D. Henderson, and D.D. Busath. 2004. Calcium block of sodium current in a model calcium channel: Cylindrical atomistic pore with glutamate side chains. Molecular Simulation 30:77-80).

Validity of the findings

The results appear very reasonable, but interpretation suffers in two ways, in my opinion. First, the glutamates are not free to follow the ions during translation through the filter, and if they were it would probably affect a point that receives a lot of emphasis from the author: that the high barrier to Ca++ at 0 Angstroms may be the source of the preference for Na+ transport, because the glutamates would probably continue to coordinate Ca++ that far into the channel. Secondly, and far more critical, the real barrier to Ca++ transport is really the barrier to leaving the filter and getting into the central cavity, a full 11 kcal/mol between 5 Ang and -5 Ang. The barrier at 0 Ang is of little importance in comparison. This is also seen in the 2-ion PMFs where the Ca++ transport faces a blind alley/dead end/box canyon at that step. Because any kinetic energy acquired entering the selectivity filter from outside the cell would be dissipated into protein and solvent kinetic energy before transport into the central cavity, the free energy of that barrier should be rate limiting. The channel passes Na+ more easily because it binds Ca++ too tightly.

Comments for the author

Why is ASP in the label in Figure 4A?
Why not remove the lipid tails from the channel? Aren't they likely to be artefacts of crystal formation?
At the end of line 3 in page 5, do you mean backbone carbonyls rather than carbonyl backbones? On the same page 15 lines up from the bottom, you have an extra "does".
It may be that the asymmetric positions of the 4 Glu side chains in eukaryotic voltage-gated calcium channels, pointed out in the early site-directed mutagenesis/electrophysiology literature of the mid-1980's, helps reduce Ca++ binding affinity to allow higher Ca++ permeability.

Reviewer 2 ·

Basic reporting

The topic is well defined, and is clearly relevant in the context of our evolving understanding of the molecular basis of ion selectivity different ion channels.

Molecular dynamics simulations, and extended analysis are based on the immediate pore domain (residues 115-221) of the NavAb structure (pdb accession code 3RVY) reported by Payandeh et al 2011 (Nature). Computational results and mechanistic conclusions are discussed in the context of experimental data from NavAb, for which data are very limited, and other bacterial Nav channels (largely NaChBac and its mutants.

Experimental design

The topic is well defined, and is clearly relevant in the context of our evolving understanding of the molecular basis of ion selectivity different ion channels.

Molecular dynamics simulations, and extended analysis are based on the immediate pore domain (residues 115-221) of the NavAb structure (pdb accession code 3RVY) reported by Payandeh et al 2011 (Nature). Computational results and mechanistic conclusions are discussed in the context of experimental data from NavAb, for which data are very limited, and other bacterial Nav channels (largely NaChBac and its mutants.

Validity of the findings

Attention is focused on the key qualitative conclusions, while clearly identifying limitations of the calculations, such as the point that polarizabilty is not included in computational models for the divalent ion calculations.

Another qualification that should be mentioned, is that the calculations are based on what Payandeh et al identify as “pre-open” state – closed at S6 bundle crossing. This provides another reason why the results and conclusions should be considered qualitative, rather than quantitative in all details. Is is clearly stated that the calculations are based only on the pore domain structure, and it is believed that selectivity filter is in an open configuration, but this leaves open the possibility that the energy contours in the selectivity filter are to some extent shaped by the presence of an “infinite” barrier, just a few Angstroms away at the S6 bundle crossing.

Comments for the author

FIGURES & LEGENDS:

Figure 5B
At -4 Angstrom, in 5A, the sum of the points is about -700 kcal/mol, but the first point in 5B is about -1 kcal/mol. How can the curves in 5B be derived as a simple sum? What am I missing?

Figures 6 & 7
I find the arrows confusing. Why not indicate the locations of the individual snapshots on the 2D surfaces by numbers 1, 2, 3, etc., and, in addition, plot the energetically favored path, through the selectivity filter for the selected ion, connecting the snapshot locations (Figs 6 & 7)? – subject, of course, to the reservations that you have stated regarding interpretation of quantitative details


DISCUSSION & CONCLUSIONS
A suggestion - The immediately following material may be better used to expand the last paragraph of the Discussion. This would leave the Conclusions to focus specifically on factors, directly identified in your calculations, which may contribute to selectivity of Na over Ca.
* * *
The closing sentence of the Conclusions, which refers to the effect of “small changes” in the protein sequence on selectivity, seems to miss, or at least downplay, a key point in the study of Yue et al., which is brought up in the last paragraph of the Discussion. As noted, all of the mutant constructs that showed substantially increased Ca conductance and / or PCa/PNa, incorporated an extra negative charge (aspartate) adjacent to the selectivity filter glutamate. Thus, the net negative charge on the filter is doubled from -4 to -8 (neglecting any changes in protonation). Surely, in functional terms, this is a major change, and should be recognized as such. Further, changing the sequence from LESWAS to LEDWAS mimics a pattern (the “ED” pair), which is highly conserved in domain II of eukaryotic Cav channels. Thus, I think it’s worth highlighting the requirement, of additional selectivity-filter negative charge, for a NaChBac mutant to approach Ca-channel like selectivity and conductance. In addition, brief comments on the two following studies would place in context the simulations of pore size and Ca binding in NavAb, alongside the structurally related eukaryotic Cav channels (Hess and Tsien, Nature, 1984 – role of Ca binding in selectivity; McLeskey and Almers, 1985 – sizing the pore).
* * *
Reviewer 3 ·

Basic reporting

Sufficient introduction and motivation has been given. Subject is very interesting. Some conclusions may not entirely be supported by the results - see general comments below.

Experimental design

Lack of evidence backing some statements and absence of statistical measures that are essential. Otherwise the methods are sufficiently described. See comments below.

Validity of the findings

It is not known if results are statistically sound. It is not known if the chosen model has captured the energetics accurately enough to demonstrate the conclusions of the study. See below.

Comments for the author

This manuscript is a continuation from a previous study of simulations of Na+ and K+ ions inside the recent NavAb voltage gated sodium channel (a simplified constrained pore-only model of that channel). Here the author has redone some of those simulations with the inclusion of a Ca2+ ion. The comparison of Na+ and Ca2+ is of course interesting for this bacterial sodium channel, as explained in the manuscript. The PMF calculations performed are of interest, but in my opinion there are several unsubstantiated claims. Some work will be needed to convince the readers that the conclusions are significant, as I explain below.

Although a new 2 ion PMF is presented, the study focuses primarily on the single ion movement through the channel, which is of some but limited relevance, and claims to explain Na/Ca selectivity in terms of barrier heights. The author sees differences in barrier heights between the ions of perhaps 10 kT units. Such an increased barrier, if the rate limiting step for permeation, would cause something like 4 orders of magnitude decrease in rate, not the 1 order seen experimentally. The question therefore is whether this is some artifact due to the model, or if this really is the mechanism for selecting against Ca2+. If this barrier really is the key to selectivity, what is it that is different in a Ca-selective channel? Regarding the barrier increase, the author does admit one cannot entirely trust the calculations due to uncertainties in the model (page 5). Should we then trust the conclusion?

Is 1 nanosecond sampling for the calculation of the free energy sufficient and why? I would need to see convergence of the free energies and reproducibility of the results. Unfortunately, a block analysis of such short simulations might give misleadingly low errors. Perhaps a better approach would be to carry out completely independent calculations to demonstrate reproducibility.

The author makes some statements that have not entirely been proved, in my opinion. E.g. On page 5, the author states that it is because of the increased charge that Ca2+ binds more strongly to the protein. This is not self-evident. E.g. On page 5 the author claims that it is one position in the plane of the side chains Glu where “selectivity of Na over K arises”, which has yet to be proved. Related, on page 2 the author claims that he has previously “been able to attribute the discrimination at this point to the inability of K+ to fit through the narrow region of the pore along with its surrounding water molecules in an optimal geometry”. Upon looking at that previous article, I see no such proof, just a snap shot from a simulation showing some idealized geometry. Unless there is quantitative proof, the wording should be modified.


Author claims a “significant drop” in the hydration of Ca below 6.5. But is this 0.3 or so water really important? What is disappointing is that the author digs up old analysis on the free energy costs of removing waters from Na, but then claims those energetics might also explain the barrier for Ca. But those calculations should be done on Ca2+ to be convincing. The statement “the barrier seen for Ca2+ seems to be due to a less than ideal combination of ion-protein and ion-water interactions” on the same page is vague.

I question the statement (page 6) that “we would expect Ca2+ to attenuate Na+ currents, but not block them”. The author should be careful not to misuse the term “block”, which of course can account for attenuated current. The author should better describe what is known about Ca block, rather than just conclude that some “small effect” supports the passing by of ions. Related on page 7, the author offers his idea that “Ca selectivity could be obtained by preventing Na from being able to pass a resident Ca ion”. Does this proposed mixed-ion mechanism relate to the experimental setup used for selectivity measurements?

In terms of the model, how much should we believe the 5 kcal/mol bigger barrier for Ca? What are the likely pitfalls in the model that were not as big an issue for the monovalent ions and is there any precedent that might allow us to judge the severity of the problems? The author’s incorrect statements such as that divalent ions are problematic in a “non-polar force field” and “Ca can be expected to polarize surrounding molecules which is an effect not considered in the present study” either reveal careless writing or a lack of familiarity with the subject. Also, the brief mention of reproduction of hydration free energies on page 7 also does not bring to the reader’s attention the more extensive testing that would be needed for accurate results.

---

## Round 0.2 · accepted · Accept

Thank you for revising the manuscript so clearly.